# Long-Term Infection and Pathogenesis in a Novel Mouse Model of Human Respiratory Syncytial Virus

**DOI:** 10.3390/v14081740

**Published:** 2022-08-09

**Authors:** Rui Xiong, Rui Fu, Yong Wu, Xi Wu, Yuan Cao, Zhe Qu, Yanwei Yang, Susu Liu, Guitao Huo, Sanlong Wang, Weijin Huang, Jianjun Lyu, Xiang Zhu, Chunnan Liang, Yihong Peng, Youchun Wang, Changfa Fan

**Affiliations:** 1National Rodent Laboratory Animal Resources Center, Institute for Laboratory Animal Resources, National Institutes for Food and Drug Control (NIFDC), Beijing 102629, China; 2Department of Microbiology & Infectious Disease Center, School of Basic Medical Sciences, Peking University Health Science Center, Beijing 100083, China; 3National Center for Safety Evaluation of Drugs, Institute for Food and Drug Safety Evaluation, National Institutes for Food and Drug Control (NIFDC), Beijing 100176, China; 4Division of HIV/AIDS and Sexually Transmitted Virus Vaccines, Institute for Biological Product Control, National Institutes for Food and Drug Control (NIFDC), Beijing 102629, China; 5Department of Pathology, InnoStar Bio-Tech Nantong Co., Ltd., Nantong 226133, China

**Keywords:** human respiratory syncytial virus, Rag2^−/−^ mice, CRISPR/Cas9, T and B cell deficiency, host immune response

## Abstract

Intensive efforts have been made to develop models of hRSV infection or disease using various animals. However, the limitations such as semi-permissiveness and short duration of infection have impeded their applications in both the pathogenesis of hRSV and therapeutics development. Here, we present a mouse model based on a *Rag2* gene knockout using CRISPR/Cas9 technology. Rag2^−/−^ mice sustained high viral loads upon intranasal inoculation with hRSV. The average peak titer rapidly reached 1 × 10^9.8^ copies/g and 1c10^6^ TCID_50_ in nasal cavity, as well as 1 × 10^8^ copies/g and 1 × 10^5^ TCID_50_ in the lungs up to 5 weeks. Mild interstitial pneumonia, severe bronchopneumonia, elevated cytokines and NK cells were seen in Rag2^−/−^ mice. A humanized monoclonal antibody showed strong antiviral activity in this animal model, implying that Rag2^−/−^ mice that support long-term stable infection are a useful tool for studying the transmission and pathogenesis of human RSV, as well as evaluating therapeutics.

## 1. Introduction

As a member of genus *Pneumovirus*, human respiratory syncytial virus (hRSV) is a negative-sense single-stranded RNA virus [1]. It was the second leading aetiology of lower respiratory infection deaths overall, with pneumococcal pneumonia the leading cause [2]. Children under 2 years of age are the main susceptible population [3], with mild upper respiratory tract infection signs, low-grade or no fever, coughing and coryza [4,5]. The severe clinical symptoms of hRSV infection include bronchiolitis especially seen in young children 2–5 months old, which can easily cause airway obstruction and hypoxia, severe wheezing, and pneumonia [6,7]. 

Although decades have passed since this pathogen was discovered in an infected chimpanzees in 1956 [8], there is still no effective and licensed hRSV vaccine has been approved [9]. Most studies are still at the experimental stage, and there was only limited success in pre-clinical studies, which is partly due to the lack of animal models that can fully recapitulate the pathogenesis of human RSV infection [10]. Notably, there are also no animal model that can support persistent infection and replication with high viral titers in vivo.

The existing hRSV animal models, including the frequently used non-human primates (NHPs) and rodents, all have specific advantages and limitations. Chimpanzees are highly susceptible to hRSV infection and allow hRSV to freely replicate in vivo [11]. Moreover, the infection titers, upper respiratory tract symptoms and histopathological changes are similar to human infants [11,12,13]. This model is unmatched by other animals, but few studies used chimpanzees as an animal model due to ethical issues, high cost and limited availability. Although it is difficult to replicate clinical signs of hRSV infection in rodents, mice and cotton rats are still widely used in the development of monoclonal antibodies to prevent severe hRSV infection in high-risk infants [14,15] and provide an animal platform for studying immune mechanism and pathogenesis [16]. However, cotton rats and mice are semi-permissible hosts for hRSV, so that high titer inoculation is required to produce effective infection, and some animals still fail to become infected [17,18]. Improved susceptibility was observed in immunocompromised cotton rats treated with cyclophosphamide [19,20]. Inspired by this study, a stably inherited *Rag2* gene knockout C57BL/6 mouse model was created using CRISPR/Cas9 technology. This novel mouse model has inherently defective T and B cells, resulting in long-term infection with hRSV and severe histopathological manifestations.

## 2. Materials and Methods

### 2.1. Animal Challenge

WT (wild-type C57BL/6) mice and Rag2^−/−^ mice (*Rag2* knockout mice) were supplied by the Institute for Laboratory Animal Resources and were raised at the barrier facility of NIFDC. Three-week-old WT (N = 24) and Rag2^−/−^ mice (N = 24) were challenged with hRSV strain Long (Reference Sequence, NC_038235.1) via the intranasal (IN) route. The mice were lightly anesthetized with pentobarbital sodium solution (15 mg/mL, 30 mg/kg) before challenging. An inoculum comprising 40 μL of hRSV virion suspension or DMEM culture medium was sucked and slowly dripped into the nostrils of the mice. 

### 2.2. Cells and Virus

HEp-2 cells were obtained from the Center for Cell Resource Conservation Research, NIFDC. Dulbecco’s modified Eagle’s medium (DMEM, Gibco) supplemented with 10% fetal bovine serum (FBS, Gibco) and 1% penicillin–streptomycin solution (PS, Gibco) was used to maintain HEp-2 cells in a 37 °C, 5% CO2 incubator. hRSV strain Long was amplified in HEp-2 cells to a titer of 1.0 × 10^6^ TCID_50_ for the challenge experiments. The virion suspension was kept at −80 °C for long-term storage.

### 2.3. Development of the Rag2^−/−^ Mouse Model

Rag2 sgRNA-oligos (Figure 1A) were designed using the CRISPR online tool crispr.mit.edu), and the linearized pT7-sgRNA vector was ligated with the annealed sgRNA fragment. In vitro sgRNA transcription and purification were conducted using the MEGA ShortScriptTM Kit (Ambion). SgRNA and Cas9 mRNA were microinjected into C57BL/6 mouse zygotes which were transplanted into pseudo-pregnant mice to obtain F0 generation mice. The offspring were genotyped using the primers Rag2-F (5′-GCCAGAAAGGCTGGCCTAA-3′) and Rag2-R (5′-CCCATGCTTTTCCCTC GACTA-3′).

### 2.4. Observation of Clinical Symptoms

The body weight of inoculated mice was measured on 6 consecutive days, and every two days after 6 dpi to 21 dpi (days post infection). The body temperature of the mice was measured by a rectal temperature measuring instrument at the same time period for 8 consecutive days, and every 1 day after 10 dpi to 18 dpi. While measuring the body temperature and weight, clinical symptoms, including respiratory signs, hunched back, or behavioral changes were observed.

### 2.5. RNA Extraction and Quantification

The mice were sacrificed using pentobarbital sodium solution (15 mg/mL, 75 mg/kg), after which the nasal tissue, trachea and lungs were collected, immediately immersed in RNAlater^®^ stabilization reagent (Invitrogen) and stored at −80 °C until further use. Total RNA was extracted from the tissues using the TRIzol reagent, and revers-transcription (RT) of hRSV RNA into cDNA was conducted using Takara’s PrimeScript RT reagent Kit with gDNA Eraser (Takara). The viral loads were determined using real-time quantitative PCR (RT-QRCR) with a reverse transcription kit containing TB green dye (Takara), using the primers hRSV-F (5′-AACGC ACCGCTAAGACAC-3′) and hRSV-R (5′-GCCATCATATTCATAG CCTCGG-3′). Data were presented as lg viral RNA copies/g tissue.

### 2.6. TCID_50_ Assay

After anesthesia, the nasal tissue, trachea and lungs were collected and weighted under aseptic conditions. Tissues were homogenized in DMEM with 1% PS to form 10% suspensions. Following centrifugation at 4 °C and 500 g for 10 min, the supernatants were passed through 0.45 μm pore-size Millipore filters and stored at −80 °C until further use. The virus titer was assayed in 96-well tissue culture plates using HEp-2 cells, which were inoculated at 37 °C and 5% CO_2_ for 5 to 6 days, as described elsewhere [19].

### 2.7. Flow Cytometry

Samples comprising 200 μL of blood were collected from the inner canthus vein of mice, and ethylene diamine tetra acetic acid (EDTA) was employed as anticoagulants. T, B and NK Cells in the peripheral blood were labeled using BioLegend fluorescently labeled antibodies, including PerCP-Cy5.5 anti-Mouse CD45 antibody, PE-Cy7 anti-Mouse CD19 antibody, APC-Cy7 anti-Mouse CD49b antibody. FACS Diva 6.0 Software was used to obtain cell data on the BD LSRII flow cytometer.

### 2.8. Cytokine Assay 

Cytokines concentrations were detected via the MSD (Meso Scale Discovery) method using the Bio-Plex Pro Mouse Cytokine 23-plex Assay kit (Bio-Rad, Berkeley, CA, USA) and quantified as pg/mL using the Bio-Plex 200 system.

### 2.9. Histopathological Analysis

The nasal tissue, trachea, and lungs were fixed in 10% neutral formalin fixation solution. The nasal tissues were decalcified using EDTA at pH 7. The tissues were trimmed, dehydrated, embedded, sectioned to about 3 μm thickness, stained with hematoxylin and eosin (H&E), and then examined using a conventional light microscope.

### 2.10. Evaluation of the In Vivo Effect of the Antibody via Bioluminescence Imaging

Groups of Rag2^−/−^ mice were intraperitoneally administrated 1 or 5 mg/kg of a monoclonal antibody (Supplied by Sino Celltech Ltd. Beijing, China) against hRSV to test the in vivo antiviral activity. Six hours after antibody administration, the mice from the antibody groups (N = 3), as well as the control group (N = 3) were challenged intranasally with 40 μL of hRSV-Fluc (1 × 10^6.5^ TCID_50_), containing a luciferase expression cassette [21]. Before imaging, mice were intraperitoneally administered the bioluminescent substrate D-Luciferin (75 mg/kg, 15 mg/mL, PerkinElmer) and then anesthetized with pentobarbital sodium intraperitoneally (75 mg/kg, 15 mg/mL). After 10 min, the bioluminescence intensity of the mice whole body or organs was measured using the IVIS-Lumina II imaging system (Xenogen). The relative intensities of the bio-optical signal were coded from red (intense) to the blue (weak) and presented quantitatively as photon flux in photons/second/cm^2^/steradian.

### 2.11. Statistical Analysis

Results of each experiment were presented as the mean values and standard deviations (SD). The data were analyzed using Prism software 9.0 (GraphPad Inc., San Diego, CA, USA). One-way ANOVA or two-way ANOVA was used to assess the significance of differences which were considered statistically significant at *p* values of less than 0.05.

## 3. Results

### 3.1. Establishment of Rag2 Knockout Mice by CRISPR/Cas9 Technology

A protein encoded by the *Rag2* gene (recombination activating gene 2) is one of the most important recombinases, which plays a key role in the initiation of V(D)J recombination during B and T cell development [22]. The *Rag2* gene generates 2 transcripts. Therefore, the common exons of the transcripts were selected, which are as close as possible to the 5′ end of the gene (Figure 1A). The predicted Rag2-sgRNA with the highest score and the lowest probability of off-target effect were selected using the MIT CRISPR online tool (Figure 1A).

The sgRNA and Cas9 mRNA were injected into 258 zygotes from C57BL/6 mice. The surviving zygotes were transplanted into 6 pseudo-pregnant mice, and 25 offspring were born (Table 1). To detect whether the Cas9 protein cleaved the sgRNA target site, the PCR amplicons were recovered and sequenced. The results confirmed that the target gene of heterozygous mouse No. 5 had a knockout 2 bp (Figure 1B,C). Positive heterozygous *Rag2* deficient mice were crossed to generate homozygous mice, which were named B6-Rag2^em5/NIFDC^, abbreviated as Rag2^−/−^ mice.

Deletion of the *Rag2* gene leads to no mature T/B lymphocytes in the peripheral blood of Rag2^−/−^ mice theoretically [23,24]. To confirm this, flow cytometry was employed to detect the presence of T, B and NK cells in the peripheral blood of homozygous Rag2^−/−^ mice. Figure 1D,E showed that the abundance of T and B cells was significantly reduced in Rag2^−/−^ mice, in line with expectations and previous reports [25]. By contrast, Rag2^−/−^ mice had slightly higher NK cell numbers than wild-type mice. In summary, we successfully established a homozygous *Rag2* knockout mouse by CRISPR/Cas9 technology, which can be used in subsequent experiments.

### 3.2. Establishment of a hRSV Infection Model Using Rag2^−/−^ Mice

Rag2^−/−^ mice were challenged intranasally with hRSV strain Long at the dose of 1 × 10^6^ TCID_50_/mL, and samples were collected at various time points (Figure 2A). There were no clinical symptoms such as respiratory signs, standing hair or arched back during the infection period up to 35 days. Infected Rag2^−/−^ mice showed no significant loss of body weight, but the average body weight of the Rag2^−/−^ mice was lower than that of both challenged and unchallenged wild-type mice from the 6th day post infection (Figure 2B). This weight fluctuation may be associated with virus infection, as was reported for Rag2^−/−^ rats infected with vaccinia virus [26]. Infected Rag2^−/−^ mice showed no significant fever, and their body temperature actually decreased on most days post infection (Figure 2C), especially on day 12.

RT-QPCR revealed the presence of hRSV virus in the nasal tissue, trachea, and lungs of Rag2^−/−^ mice. The viral loads on the 4th day reached about 1 × 10^7^–1 × 10^9^ copies/g in lungs and nasal tissue, implying a rapid virus replication occurring in Rag2^−/−^ mice (Figure 2D). The viral load continued to rise until the peak of 1 × 10^9.8^ copies/g (mean) on 11 dpi and was detectable in the nasal tissue up to 35 dpi. In the lungs, the viral load reached the peak of 1 × 10^9^ copies/g on 11 dpi and remained at this level up to 35 dpi. These results indicated that there was a delay of viral replication in the trachea, which reached the highest viral titer of 1 × 10^7^ copies/g approximately on day 20 post infection. However, it remained at a high titer during the infection (Figure 2D). By contrast, wild-type mice could only support a shorter, burst of viral replication at lower titers, followed by rapid viral clearance from both the upper and lower respiratory tract, consistent with previous reports [16]. Hence, there was a significant difference in susceptibility between Rag2^−/−^ and wild-type mice, as expected.

In addition to RT-QPCR, homogenates of nasal tissue, trachea and lungs were used to determine the titer of live hRSV (Figure 2D). The results showed that the change trend of live hRSV virus titer in the respiratory tract was basically consistent with the total viral copies detected by RT-QPCR detections, with high and stable hRSV replication in nasal tissue, trachea, and lungs, with titers of 1 × 10^4^–1 × 10^6.5^ TCID_50_ lasting for 5 weeks, until the end of the experiments. By contrast, the wild-type mice had detectable viable virions in the trachea and lung tissues collected on 4th day, after which they could not be detected in all the mice, indicating rapid clearance.

### 3.3. Histopathological Findings of a hRSV Infection Model Using Rag2^−/−^ Mice

The histopathological changes in susceptible organs were consistently observed. On 4 dpi, no obvious changes were seen in the nasal cavity and trachea of Rag2^−/−^ mice and wild-type mice (Figure 3B,F), while minimal mononuclear cell infiltration around the blood vessels were seen in lung tissues. On 11 dpi, the transitional epithelium in the nasal cavity of Rag2^−/−^ mice showed degeneration and inflammatory cell infiltration (Figure 3D). At the same time, obvious histopathological changes appeared in the lower respiratory tract of partial some Rag2^−/−^ mice (Figure 3H–P). Although some areas of the lung tissue were basically normal (Figure 3I), other areas showed severe lesions (Figure 3H), such as dilated and severely compressed bronchi and alveolar spaces (Figure 3K), accumulation inflammatory cells (predominantly neutrophils) in the bronchial lumen and alveolar spaces (Figure 3J,L,O), and enlargement of the alveolar septum (Figure 3P). Most of the neutrophils located in the center of the bronchial lumen were dead and degraded (Figure 3M), while cells at the edges still survived (Figure 3L). It seemed that the histopathological changes recovered on 35 dpi, and no lesion were seen in tissues sampled at this time. The infected and mock-treated wild-type mice showed no inflammatory cell infiltration or other pathological changes. In all, the results indicated that hRSV infection could cause mild interstitial pneumonia, severe bronchopneumonia and suppurative bronchitis in Rag2^−/−^ mice.

### 3.4. hRSV Infection Causing Upregulation of NK Cells, Serum Cytokines and Chemokines

The T, B, NK and NKT cell counts in peripheral blood were determined by flow cytometry on 0, 4, 20, and 35 dpi (Figure 4A). Since the *Rag2* gene was knocked out, the T and B cell counts have no obvious fluctuation in Rag2^−/−^ mice. However, the NK cell counts in the peripheral blood of Rag2^−/−^ mice significantly increased (more than 20%) after infection, and reached a plateau at 4 dpi, where it stayed until 35 dpi (Figure 4B). The NKT cell count also increased in the early stage of infection and then decreased.

In addition to detecting immune cells, a total of 23 cytokines in the sera of Rag2^−/−^ and wild-type infected mice were quantified using the MSD method [27]. Most expression of cytokines including IFNG did not change significantly after hRSV infection in Rag2^−/−^ mice and wild-type mice (Figure 4C). However, the concentrations of IL1A, IL1B, IL6, CSF3(G-CSF), CSF2(GM-CSF), and CCL2(MCP-1) in infected Rag2^−/−^ mice increased at each time-point, while there were almost no changes in wild-type mice (Figure 4D). The concentrations of IL-1a, IL-1b and GM-CSF in Rag2^−/−^ mice remained at the same high level up to 20 days of infection, whereas the expression of IL-6 reached a peak on the 4th day, and then decreased to the initial level.

### 3.5. A BLI Rag2^−/−^ Mouse Model Evaluating the In Vivo Antiviral Activity of a Humanized Monoclonal Antibody

To establish a bioluminescence luminescence imaging (BLI) animal model, Rag2^−/−^ mice were infected with hRSV-Fluc intranasally (Figure 5A). Obvious light was seen in the nasal cavity of the mice after 4 days of inoculation in the PBS group (Figure 5B) and the bioluminescence signal remained at a high level up to 11 dpi (Figure 5C,D).

To evaluate the in vivo antiviral activity of humanized monoclonal antibody against hRSV, two dosages (1 and 5 mg/kg) of the antibody were administered to Rag2^−/−^ mice. The results showed that no obvious luminescent signal was detected in the whole body of mice until 11 dpi. The nasal tissue, trachea and lungs were taken for tissue imaging, and only the nasal tissue and lungs of the mice in the control group (mock-treated with PBS) had obvious fluorescence signals (Figure 5E,F), while the corresponding organs from the mice administered with 1 or 5 mg/kg of the antibody showed no light signals, indicating that even a lower dose of the antibody could effectively inhibit viral infection and replication in Rag2^−/−^ mice and the humanized monoclonal antibody had strong antiviral activity.

## 4. Discussion

Because of the semi-permissive host [10], the virus cannot persistently replicate at high titer in the preexisting animal models, resulting in rapid clearance [16,28] and inability to evaluate the long-lasting antiviral activity of drugs in vivo. Previous reports showed that rats dosed with immunosuppressor showed increased permission of hRSV [19,20,29]. Obviously, to establish such drug-induced model needs complicated operations and accompanied with poor repeatability. In this study, CRISPR/Cas9 technology was used to delete *Rag2* gene in C57BL/6 mice to establish a genetically inherited immunodeficient mouse model, which was expected to support long-term hRSV infection.

Rag2^−/−^ mice supported efficient viral replication of hRSV in respiratory organs, including the nasal cavity, trachea, and lungs, which is consistent with both upper and lower respiratory tract infection of clinical patients [30]. The viral replication level was very high in nasal cavity and lungs (Figure 2). Notably, such an efficient infection also had a long duration, and maintained a stable infectious state, so that even at the end of the experiments on the 35th day, no decrease of viral titer was observed (Figure 2D). It is possible that the infection duration in Rag2^−/−^ mice would last longer than the 6 weeks reported previously in cotton rats [19,20]. In addition, this Rag2^−/−^ also showed typical pathological changes (Figure 3) and host immune responses (Figure 4), indicating Rag2^−/−^ mice had clear advantages over existing rodent models of hRSV.

In addition to the symptoms of pneumonia such as inflammatory cell infiltration, and thickened alveolar septum, severely suppurative bronchitis and bronchopneumonia associated with increased neutrophil infiltration were seen in Rag2^−/−^ mice. Neutrophils are the major leukocytes that infiltrate the airways [31] and studies have shown that longer hRSV infection lasts, leads to higher neutrophil infiltration of the airways and enhances the ability of hRSV to damage and isolate airway epithelial cells [32]. It is worth mentioning that hRSV is one of the causative pathogens of viral sepsis [33], which was often found in immunocompromised hosts infected by respiratory virus [34]. In this study, Rag2^−/−^ mice kept immune suppression stage with no functional T and B cells (Figure 1) and suffered from a high level of viral infection (Figure 2). These characteristics are conditions that can allow hRSV-infected Rag2^−/−^ mice to develop viral sepsis. If the infected Rag2^−/−^ mice were observed for longer, more severe clinical symptoms such as sepsis might be found. Overall, the observed pathological changes indicated the potential of Rag2^−/−^ mice as a model for future studies focusing on the pathology of viral infection.

In suckling mice, NK cells will appear quickly in the lungs and BAL after infection [35]. The activity of NK cells reaches its maximum on the 3rd day and resolves on the 8th day [36]. Here, we observed that the number of NK cells increased quickly upon the hRSV infection (Figure 3) in Rag2^−/−^ mice, which have a severe lack of T and B cells, indicating compensation for adaptive immune deficiency. Thus, NK cells may participate in virus clearance in the early stage of hRSV infection. However, NK cells may not the main cells terminating hRSV infection since there was high-level replication from the beginning to the end (Figure 3).

According to previous reports, humans infected with hRSV have high levels of pro-inflammatory cytokines, such as TNF, IL6, IL1A, and CXC/CC, as well as chemokines such as CXCL8, CCL3, CCL2 and CCL5 [37], IFNG, IL4, IL5, IL10, IL9 and IL17A [38], and also the colony stimulating factor CSF3 [39]. In this study, the expression of IL1A, IL1B, IL6, CSF3, CSF2, and CCL2 was up-regulated, which was similar to that of clinical patients [40,41,42]. In this study, the cytokines that were altered with hRSV infection could be secreted by somatic cells rather than immune cells [43,44,45,46], and these cytokines may correlate with virus infection, replication, and pathogenesis. Infection of respiratory epithelial cells by hRSV [47] results in high expression of pro-inflammatory factors such as IL6, which may play a relevant role in the pathogenesis of hRSV in the lungs of infected infants [48], as well as Rag2^−/−^ mice. It is worth noting that T and B cell deficiency may result in the lack of secretion of some cytokines, so that the innate immunity and adaptive immune responses cannot mount an effective antiviral defense. As a result, the positive feedback loop in which immune cells secrete large amounts of cytokines, which in turn stimulates immune cells during the immune response may be blocked. This may be a possible explanation why the Rag2^−/−^ mice can support long-term, high level hRSV infection, but fail to show obvious clinical symptoms as in humans infected with hRSV.

No vaccines against hRSV have been registered to date, and the development of therapeutic drugs, including antibodies and chemicals, is urgently needed. Here, we presented a visual in vivo efficacy evaluation model based on the highly susceptible Rag2^−/−^ mouse, which will contribute to the development of therapeutics in the future. The limitation of the present mouse model is that it cannot be used to evaluate the in vivo efficacy of vaccines, which can be overcome by developing an immunocompetent mouse expressing human RSV receptors.

## Figures and Tables

**Figure 1 viruses-14-01740-f001:**
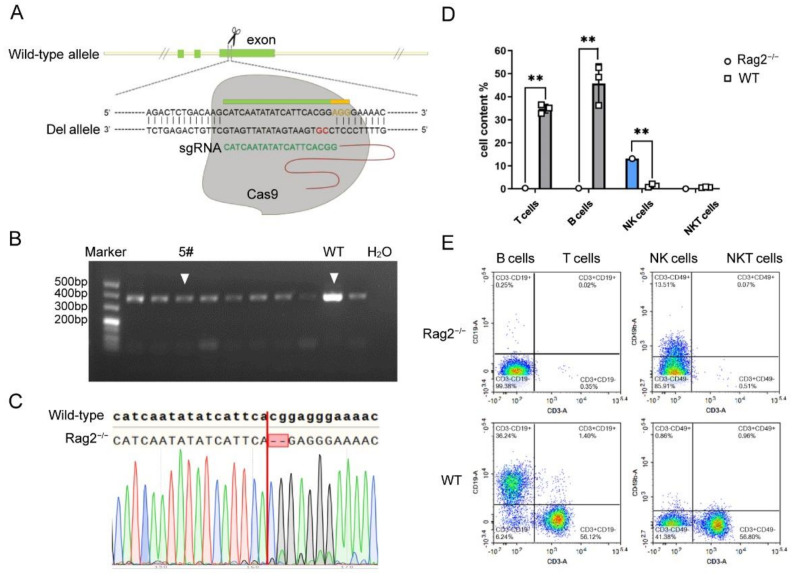
Establishment of *Rag2* knockout mice by CRISPR/Cas9 technology. (**A**) Schematic illustration of the strategy for the generation of *Rag2* knockout mice. (**B**) PCR genotyping to screen founder mouse, the expected PCR amplicon length was 362 bp. (**C**) PCR amplicons of candidate founder mice were sequenced, and two bases (CG) were deleted. (**D**) Flow cytometry test analysis of T, B, and NK cells in the peripheral blood of wild-type C57BL/6 mice and homozygous Rag2^−/−^ mice. The corresponding cytometric dot-plots are shown in figure (**E**). The statistical significance was evaluated using the unpaired *t*-tests, ** *p* < 0.01.

**Figure 2 viruses-14-01740-f002:**
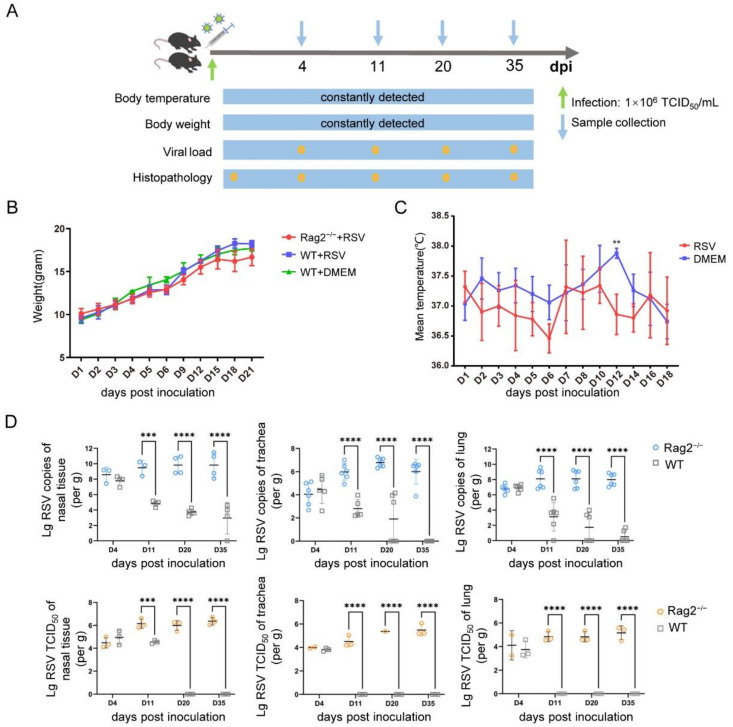
Establishment of an hRSV infection model using Rag2^−/−^ mice challenged intranasally. (**A**) Challenging and sampling schedules for establishment of the hRSV infectious model. (**B**) Body weight of Rag2^−/−^ mice (N = 4) and wild-type mice (N = 4) during the 21-day observation period post infection. (**C**) Rectal temperature during the 18-day observation period after challenging Rag2^−/−^ mice with hRSV (N = 5) or mock-challenge with DMEM (N = 5). (**D**) Total hRSV copies and viable hRSV loads in the target organs on 4, 11, 20, and 35 dpi according to RT-QPCR and TCID_50_. The statistical significance was evaluated by two-way ANOVA. ** *p* < 0.01, *** *p* < 0.001, **** *p* < 0.0001.

**Figure 3 viruses-14-01740-f003:**
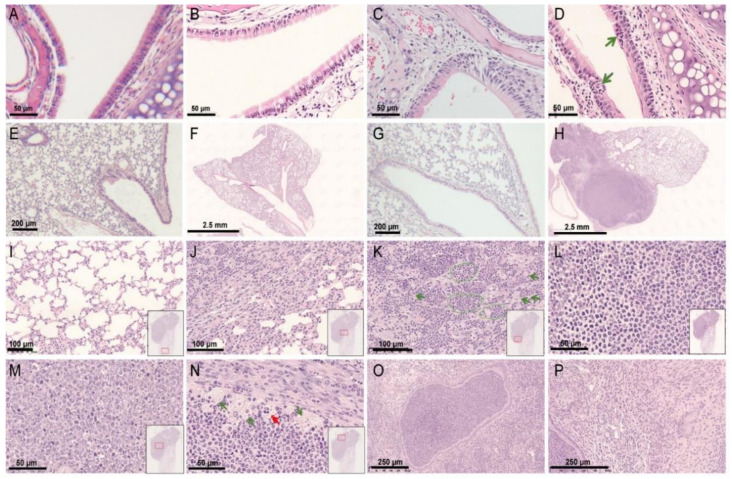
Histopathological findings in the respiratory tract of hRSV-infected Rag2^−/−^ mice. Hematoxylin and eosin staining showed no obvious histopathological changes in the nasal tissue of wild-type mice inoculated with DMEM (**A**) or hRSV (**B**). (**C**) No obvious histopathological changes in the nasal tissue of Rag2^−/−^ mice inoculated with DMEM. (**D**) Degeneration of transitional epithelium and inflammatory cell infiltration observed in the nasal cavity of Rag2^−/−^ mice challenged with hRSV on 11 dpi. No obvious histopathological changes in the lung tissue of wild-type mice inoculated with DMEM (**E**) or hRSV (**F**) and Rag2^−/−^ mice with DMEM (**G**). (**H**) Part of the lung tissue of Rag2^−/−^ mice was basically normal (upper right), but severe suppurative bronchitis and bronchopneumonia were found in the left part of the lung on 11 dpi. (**I**) The basically normal area of the lung tissue of Rag2^−/−^ mice, indicated by the red rectangle in the insert in Figure 3I, similarly below. (**J**) The part of the lung tissue with infiltration of neutrophils. (**K**) Some alveoli (indicated by green circles) were filled with neutrophils, and some alveoli (indicated by green arrows) were severely compressed and deformed. (**L**) The bronchial lumen was filled with inflammatory cells (predominantly neutrophils). (**M**) There was a large number of degenerating and necrotic neutrophils in the central area of the bronchial lumen. (**N**) Enlarged and necrotic cells at the edge of the bronchial lumen (indicated by green and red arrows respectively). (**O**) Bronchial lumen filled with neutrophils, and the bronchial epithelium exists without obvious degeneration and shedding. (**P**) Enlargement of alveolar septa and interstitial inflammation including infiltration of monocytes and neutrophils in the lungs of Rag2^−/−^ mice.

**Figure 4 viruses-14-01740-f004:**
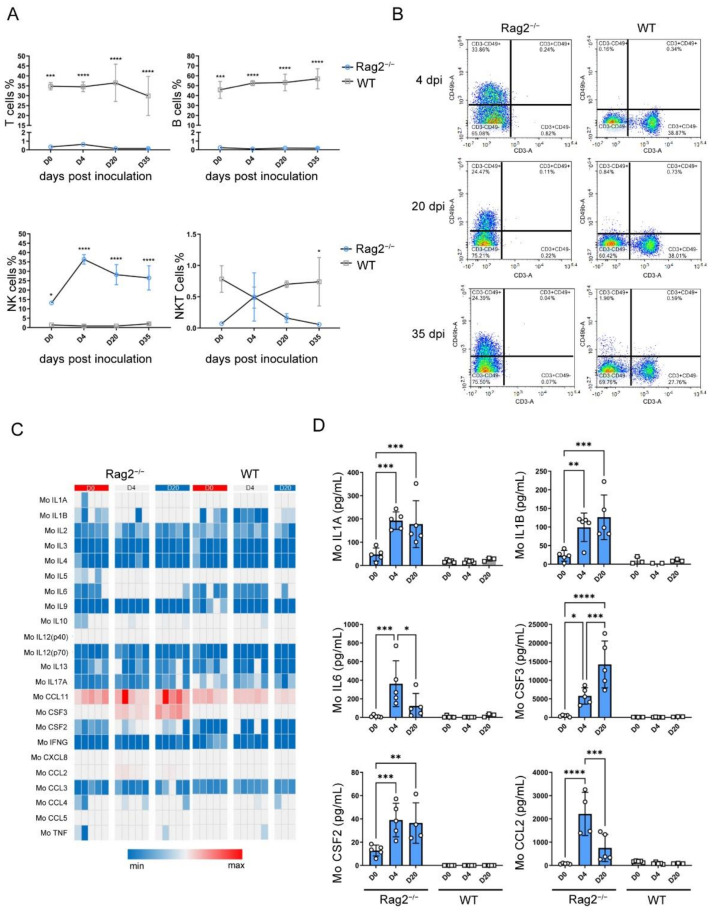
Abundance of immune cells and inflammatory response in Rag2^−/−^ mice during hRSV infection. (**A**) Comparison of the contents of T, B, NK and NKT cells in Rag2^−/−^ and wild-type mice on various time points following hRSV infection. The statistical significance was evaluated using two-way ANOVA. **p* < 0.05, ****p* < 0.001, *****p* < 0.0001 (**B**) The abundance of NK cells in a representative homozygous Rag2^−/−^ mouse at 3 time points after infection. A total of 23 cytokines were quantified using the Bio-Plex Pro Mouse Cytokine 23-plex Assay kit. (**C**) Heat map of peripheral blood cytokine concentration (pg/mL) of Rag2^−/−^ and wild-type mice on 0, 4, and 21 dpi after infection. The relative concentration of 23 cytokines were showed from the blue (low) to red (high). (**D**) The expression of IL1A, IL1B, IL6, CSF3(G-CSF), CSF2(GM-CSF), and CCL2(MCP-1) at each time point of hRSV infection. The statistical significance was evaluated using two-way ANOVA. * *p* < 0. 05, ** *p* < 0. 01, *** *p* < 0.001, **** *p* < 0.0001.

**Figure 5 viruses-14-01740-f005:**
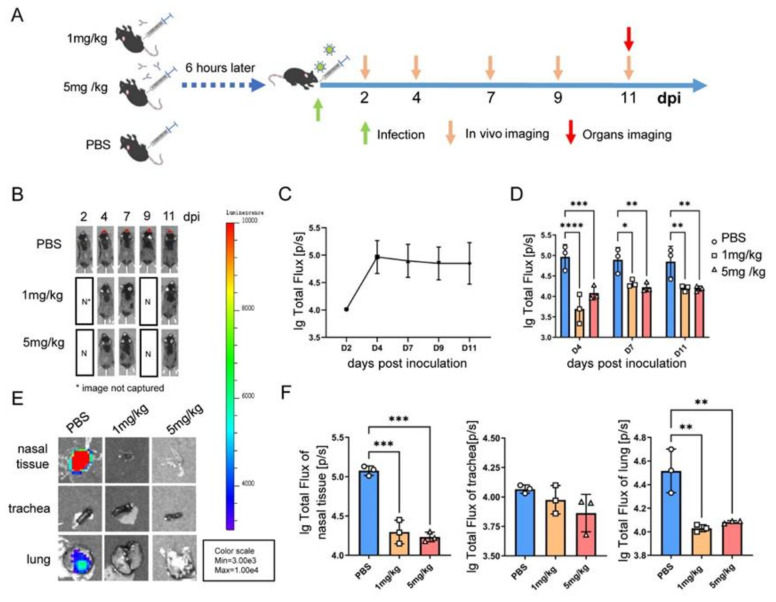
Establishment of a BLI model and evaluation of the in vivo antiviral activity of humanized monoclonal antibody. (**A**) Experimental schedule for the establishment and evaluation of BLI model based on Rag2^−/−^ mice. Two dosage of antibody groups (1 and 5 mg/kg) were administered intraperitoneally 6 h before the hRSV-Fluc challenge, the control group was administered with PBS. (**B**) All Rag2^−/−^ mice were intranasally infected with hRSV-Fluc (1 × 10^6.5^ TCID_50_), and the Bioluminescent images of the live micewere measured using the IVIS-Lumina II imaging system on 2, 4, 7, 9, and 11 dpi. Relative bioluminescence intensity was shown in pseudocolor, with red representing the strongest and blue representing the weakest photon fluxes. The bioluminescence intensity of control group mice at 5 time points (**C**) and comparison between 3 groups at 3 time points (**D**) after infection are shown as means ± deviation. The statistical significance was evaluated using two-way ANOVA. * *p* < 0. 05, ** *p* < 0. 01, *** *p* < 0.001, **** *p* < 0.0001 (**E**,**F**) Organs were examined for Fluc expression using BLI on 11 dpi. The statistical significance was evaluated using one-way ANOVA. ** *p* < 0. 01, *** *p* < 0.001.

**Table 1 viruses-14-01740-t001:** Generation of Rag2^−/−^ mice by CRISPR/Cas9 technology.

Injection Batches	Zygotes Injected	Zygotes Transferred	Newborns	F0 Mice
No. 1	114	53	9	1
No. 2	144	82	16	0

## Data Availability

Data available upon request.

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
