# Peer review of "Long-Term Infection and Pathogenesis in a Novel Mouse Model of Human Respiratory Syncytial Virus"

_viruses, 2022, doi:10.3390/v14081740_

Round 1
Reviewer 1 Report
This study provided an immunodeficient mice model for hRSV infection. Although this model showed viral infection and replication (expected) with no clinic symptoms (unfortunately), it can be used to test antiviral drugs in vivo.
Specific Points:
1. As mentioned, Rag-/- mice were established and reported by others [line 178, ref 26, 27], to make paper more concise, Fig.1 can be put as supplementary data.
2. Fig.3 should show data from control (WT, and Rag2-/-, without challenge) and test groups (challenged with hRSV) side-by-side for comparison and easy understanding.
3. Fig.1 showed big loss of B/T cells, and Fig.4 showed significant cytokine increase in Rag-/- mice with hRSV infection. Where were the cytokine from? Were they from NK cells or infected somatic cells? Please provide such data in the revision.
4. The authors cited too many review papers. Instead, more original research articles should be cited.
5. More detailed figure legends should be provided for readers' followup.
Reviewer 2 Report
Human respiratory syncytial virus is particularly interesting because today, during the COVID-19 pandemic. The authors are absolutely correct saying that it occupies a special place among respiratory viruses, continuing to circulate in the same way as rhinoviruses, while other respiratory viruses are suppressed by the SARS-CoV-2. I believe that the knockout model of RS-virus infection which the authors have developed is extremely useful and will find its application in a variety of fields, from classical virology to the selection of promising chemotherapy drugs.
Point 1: The article title is too general and vague. If the authors first developed such a wonderful model of RS-virus infection, this must be reflected in the title!
Point 2: Line 35-36. Please, indicate which infection is on the first place.
Point 3: Line 48, 51, 137, 294, 305, 316, 371, 373. Please, italicize “in vivo”
Point 4: Line 68. I guess “ethics committee” should be capitalized.
Point 5: Line 88. Please, italicize “in vitro”
Point 6: Text on line 155-161 should be transferred immediately after 3.1 (line 153). Line 162-168 should be placed just below Figure 1.
Point 7: Figure 2B. I would hardly recommend to rearrange the figure and recalculate data in %, where 100% is the weight of mice at the first day of measuring.
Point 8: Figure 2C. The picture of body temperature dynamics changes will be much clearer in graphic form, not in bars.
Round 2
Reviewer 1 Report
The authors have addressed the questions raised by the reviewer.